# Fingerprints of supersymmetric spin and charge dynamics observed by inelastic neutron scattering

Björn Wehinger [1,2,10] ✉, Franco T. Lisandrini [3], Noam Kestin[1], Pierre Bouillot[1], Simon Ward [2,11], Benedikt Thielemann[2], Robert Bewley[4], Martin Boehm [5], Daniel Biner[6], Karl W. Krämer [6], Bruce Normand[7,8], Thierry Giamarchi [1], Corinna Kollath[3], Andreas M. Läuchli[7,8] & Christian Rüegg[1,7,8,9]

Supersymmetry is an algebraic property of a quantum Hamiltonian that, by giving every boson a fermionic superpartner and vice versa, may underpin physics beyond the Standard Model. Fractional bosonic and fermionic quasi-particles are familiar in condensed matter, as in the spin and charge excitations of the *t-J* model describing electron dynamics in one-dimensional materials, but this type of symmetry is almost unknown. However, the triplet excitations of a quantum spin ladder in an applied magnetic field provide a super-symmetric analogue of the *t-J* chain. Here we perform neutron spectroscopy on the spin-ladder compounds $(C_5D_{12}N)_2CuBr_4$ and $(C_5D_{12}N)_2CuCl_4$ over a range of applied fields and temperatures, and apply matrix-product-state methods to the ladder and equivalent chain models. From the momentum-resolved dynamics of a single charge-like excitation in a bath of fractional spins, we find essential differences in thermal broadening between the supersymmetric and non-supersymmetric sectors. The persistence of a strict zone-centre pole at all temperatures constitutes an observable consequence of supersymmetry that marks the beginning of supersymmetric studies in experimental condensed matter.

Supersymmetry is a property of a quantum Hamiltonian that any bosonic particle has a dual fermionic superpartner and vice versa[1]. In quantum field theory, it sets additional constraints on the dynamics that can allow analytical progress at strong coupling[2]. As a dynamical symmetry of composite particles, it is used in nuclear physics to classify multiplets of heavy nuclei[3,4]. Despite the aesthetic appeal of the possibility that it is a fundamental symmetry of nature, experimental

evidence to date suggests that it would be severely broken at presently accessible energy scales[2,5]. In condensed matter, the concept that strong, many-body correlations can produce fractional bosonic and fermionic collective states is well established in low dimensions, but possible supersymmetries between these states are rarely invoked. Beyond some theoretical analysis of models constructed with explicit supersymmetry,[6,7] its leading applications have been in

[1]Department of Quantum Matter Physics, University of Geneva, Geneva, Switzerland. [2]PSI Center for Neutron and Muon Sciences, Paul Scherrer Institute, Villigen-PSI, Switzerland. [3]Physikalisches Institut, University of Bonn, Bonn, Germany. [4]ISIS Facility, Rutherford Appleton Laboratory, Oxford, United Kingdom. [5]Institut Laue-Langevin, 38042 Grenoble, France. [6]Department of Chemistry, Biochemistry and Pharmaceutical Sciences, University of Bern, Bern, Switzerland. [7]PSI Center for Scientific Computing, Theory and Data, Paul Scherrer Institute, Villigen-PSI, Switzerland. [8]Institute of Physics, Ecole Polytechnique Fédérale de Lausanne (EPFL), Lausanne, Switzerland. [9]Institute for Quantum Electronics, ETH Zurich, Hönggerberg, Switzerland. [10]Present address: European Synchrotron Radiation Facility, Grenoble, France. [11]Present address: Novo Nordisk A/S, Research and Early Development, Måløv, Denmark. ✉e-mail: bjorn.wehinger@esrf.fr

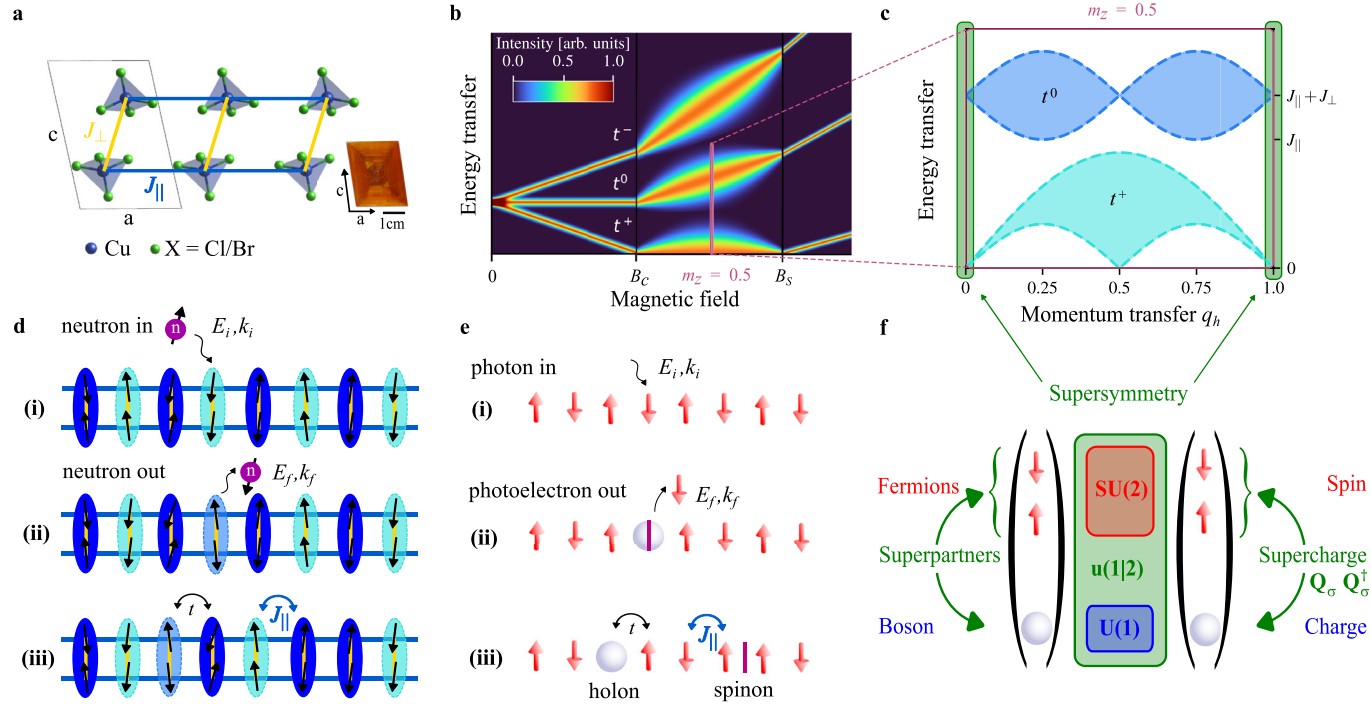

**Fig. 1 | Realization of a supersymmetric *t-J* chain from a spin ladder in an applied magnetic field. a** Schematic representation of the structure of $(C_5D_{12}N)_2CuBr_4$ (BPCB) and $(C_5D_{12}N)_2CuCl_4$ (BPCC) projected on the *ac* plane, accompanied by a photograph of one BPCC crystal. $Cu^{2+}$ ions (blue) form the ladder and halide ($Br^-$ or $Cl^-$) ions (green) the superexchange paths yielding the interactions $J_\perp$ (yellow) and $J_\parallel$ (blue). The piperidinium ions are omitted for clarity. **b** Evolution of triplet excitation branches in a strong-rung spin-1/2 ladder in a field, $B^z$. For $B_c \leq B^z \leq B_s$, the spin degrees of freedom fractionalize and the response becomes finite over a continuum of energies in each of the triplet branches $t^+$, $t^0$ and $t^-$ (the illustration represents an integral over all $q_h$ in panel **c**). **c** Many-body physics in the spin ladder at zero temperature and half-magnetization ($m^z = 0.5$). The $t^+$ sector gives rise to the turquoise two-spinon continuum and the $t^0$ sector the blue spinon-holon continuum, which is described by an adapted *t-J* model. Supersymmetry between the spinon and holon is manifest at wavevector $q_h = 0$ (green).

**d** Schematic representations of (i) singlet ($|s\rangle = \frac{1}{\sqrt{2}}(|\uparrow\downarrow\rangle - |\downarrow\uparrow\rangle)$) and triplet ($|t^+\rangle = |\uparrow\uparrow\rangle$) states on the ladder rungs in the ground manifold created by the applied field; (ii) creation of a "middle triplon" ($|t^0\rangle = \frac{1}{\sqrt{2}}(|\uparrow\downarrow\rangle + |\downarrow\uparrow\rangle)$) by a neutron-mediated spin excitation on a rung; (iii) propagation of the $|t^0\rangle$ excitation and rearrangement of the $|s\rangle$-$|t^+\rangle$ configuration. **e** Representation of these states in the basis of pseuospins (red arrows) $|\tilde\uparrow\rangle \equiv |t^+\rangle$ and $|\tilde\downarrow\rangle \equiv |s\rangle$ on a chain (i), where flipping one spin creates two freely mobile domain walls (spinons). Exciting a $|t^0\rangle$ by INS is analogous to a photoemission process, where the injected "hole" is accompanied by one spinon (ii), both of which can propagate separately (iii).
**f** Representation of supersymmetry as an extended symmetry connecting the three site-basis states of the pure *t-J* model at $J = 2t$: the additional duality between the fermionic spins (with SU(2) spin symmetry) and the bosonic hole (with U(1) charge symmetry), which makes them superpartners, is encoded in the supercharge operators $Q_\sigma$ and $Q_\sigma^\dagger$.

supersymmetric field theories relevant to critical phenomena[8–10] and topology[11]. To our knowledge even a simple supersymmetric Hamiltonian has yet to be engineered using ultracold atoms, despite extant theoretical suggestions[12–14], but this has been achieved in a recent experiment using trapped ions[15].

The *t-J* model is a paradigm for describing charged particles interacting strongly with an environment of correlated quantum spins, and is thought by some to capture the fundamental ingredients responsible for Mott physics[16] and the enduring mysteries of cuprate superconductivity.[17] Unlike the closely related Hubbard model[18], in one dimension the *t-J* model is integrable and exactly soluble (by Bethe-Ansatz methods) only at one ratio of the hole-hopping and spin-interaction parameters, $2t/J = 1$[19–21]. At this ratio the model also has explicit supersymmetry: the two fermions representing the SU(2)-symmetric spin sector and the boson representing the U(1) charge sector are superpartners, raising the symmetry to u(1|2). Despite the many intriguing consequences of supersymmetry, the field of correlated quantum matter has retained a strong focus on the parameter regime $t \gg J$, which applies in almost all materials offering real and separable charge and spin degrees of freedom.

In experiment, the physics of a single charge in a magnetic environment has been probed by photoemission spectroscopy, and

signatures of spin-charge separation have been reported in 1D metals[22,23], insulators[24–26] and cold-atom systems,[27] as has spin-orbital separation in a 1D insulator.[28] In this context, low-dimensional quantum magnets offer a wide variety of exotic physical properties, generically low (meV) energy scales accessible to laboratory magnetic fields and high-intensity, high-resolution measurements of the full spectral function by modern inelastic neutron scattering (INS) spectrometers. Magnetic-field control of the quasiparticle density in the two-leg quantum spin ladder, represented in Fig. 1a–c, allowed the quantitative testing of phenomena such as field-induced Bose-Einstein condensation and the formation of the spin Tomonaga-Luttinger liquid (TLL), which exhibits the fractionalization of magnon excitations into deconfined spinons[29–32]. It has been pointed out that, in parallel, the two-leg ladder in a field also provides a rather faithful realization of a single hole in the *t-J* model, with one excitation in the middle Zeeman-split triplet branch ($t^0$) playing the role of this hole[33], as represented in Fig. 1d–e. Here we point out that, in addition, this ladder-based *t-J* model is supersymmetric.

In this Article, we investigate the consequences of supersymmetry in the ladder-derived *t-J* model, and in the process take the first steps in studying supersymmetry as an experimental science. The basic tenets of supersymmetry as an extended symmetry of the *t-J* basis are very simply stated, but to understand their manifestations in

a condensed-matter system we proceed from ladder materials through INS experiments and numerical many-body calculations. Specifically, we perform high-resolution INS measurements of two quantum spin-ladder materials, using the applied magnetic field and the temperature as control parameters. We focus on the spectral functions of the middle ($t^0$) triplet branch in the two parity sectors of the ladder to highlight the presence or absence of supersymmetry between the charge and spin fractions of the electron in the equivalent $t$-$J$ chain. For a quantitative analysis of our results, we perform zero- and finite-temperature matrix-product-states (MPS) calculations both for the ladder and for selected $t$-$J$ models. We show how the presence of an exact pole in the ladder spectrum has consequences in the $t$-$J$ model that allow us to demonstrate observable fingerprints of supersymmetry in condensed matter.

## Results

### Supersymmetry in a spin ladder

To realize supersymmetric physics we consider the deuterated compounds bis-piperidinium copper(II) bromide (BPCB) and chloride (BPCC), which form excellent two-leg quantum spin ladders. The $Cu^{2+}$ ions carry localized $S = 1/2$ moments and through double halide bridges form spin dimers constituting the ladder rungs (yellow lines in Fig. 1a), while single halide bridges form ladder legs (blue lines) along the crystalline $a$ axis[34]. Neighboring ladders are separated by the large, organic piperidinium ions, $(C_5D_{12}N)^+$. Crystal growth and structure are described in the Methods section and in Supplementary Note 1[35]. The microscopic Hamiltonian describing the compound in an applied magnetic field is

$$H = J_\perp \sum_i \mathbf{S}_{i,1} \cdot \mathbf{S}_{i,2} + J_\parallel \sum_{i,\eta} \mathbf{S}_{i,\eta} \cdot \mathbf{S}_{i+1,\eta} - b^z \sum_{i,\eta} S^z_{i,\eta}, \quad (1)$$

in which the spin operator $\mathbf{S}_{i,\eta}$ acts at site $i$ of leg $\eta \in \{1, 2\}$ in the ladder and $b^z = g\mu_B B^z$ with $\mu_B = 0.672$ K/T the Bohr magneton, $g$ the gyromagnetic factor and $B^z$ the applied magnetic field. The antiferromagnetic Heisenberg interaction parameters determined by INS for BPCB are $J_\perp = 12.6$ K on the ladder rung and $J_\parallel = 3.55$ K on the leg, with $g = 2.28$ for a field parallel to the $b$ axis[34], while for BPCC $J_\perp = 3.42$ K, $J_\parallel = 1.34$ K and $g = 2.26$[32]. Thus both materials are strong-rung ladders with a similar ratio $J_\perp/J_\parallel$, which is well adapted for separating the triplet excitation sectors by energy in an applied field.

The energy scales of $J_\perp$ and $J_\parallel$ in both materials are well suited for high-precision INS measurements, including in fields up to saturation ($B_s = 13.79$ T in BPCB and 4.07 T in BPCC). The lower energy scale in BPCC facilitates working at finite temperatures without incurring background comparison problems. The interladder coupling $J' \lesssim 0.002\, J_\perp$ in both cases[32,36] is sufficiently small that we focus on the properties of isolated ladders [Eq. (1)]. INS experiments were performed on the triple-axis spectrometer ThALES at the Institut Laue Langevin (ILL) and on the time-of-flight spectrometer LET[37] at the ISIS pulsed neutron source (Methods section). The former were prioritized to probe ladder physics at different applied fields and the latter to probe the spectral response with optimal energy and momentum resolution. The elegant parity selectivity of the ladder geometry allows a complete separation of singlet-triplet excitation processes, which appear in the antisymmetric rung momentum sector ($q_\perp = \pi$), and triplet-triplet processes, which appear in the symmetric sector ($q_\perp = 0$)[32,33]; a detailed discussion is provided in Supplementary Note 2[35].

As noted above, the physics of a single doped hole in a $t$-$J$ chain appears in the two-leg spin ladder when describing the response of the $t^0$ sector, depicted in blue in Fig. 1c[33]. As Fig. 1d−e represent schematically, one $t^0$ excitation appears as a hole $|0\rangle \equiv |t^0\rangle$ in the background of pseudospins $|\tilde\uparrow\rangle$ and $|\tilde\downarrow\rangle$ representing the $|s\rangle$-$|t^+\rangle$ ground-state

condensate at fields $B_c \le B \le B_s$. For the Hamiltonian of Eq. (1), this $t$-$J$-chain model takes the form

$$\begin{aligned} H_{tJl} &= t \sum_i (c^\dagger_{i\tilde\uparrow} c_{i+1\tilde\uparrow} + c^\dagger_{i\tilde\downarrow} c_{i+1\tilde\downarrow} + \text{H.c.}) \\ &+ J_1 \sum_i (n_{ih} n_{i+1} + n_i n_{i+1h}) + \mu \sum_i n_{ih} + H_{XXZ}, \end{aligned} \quad (2)$$

where $t = \frac{1}{2}(J_\parallel, J_1 = -\frac{1}{4}J_\parallel$ and $\mu = \frac{1}{2}(b^z + J_\perp)$. $c^{(\dagger)}_{i\sigma}$ annihilates (creates) a fermion at site $i$ with pseudospin $\sigma = \tilde\uparrow, \tilde\downarrow$, $n_i$ is the fermion number operator and $n_{ih}$ is the hole number operator. In the absence of a hole

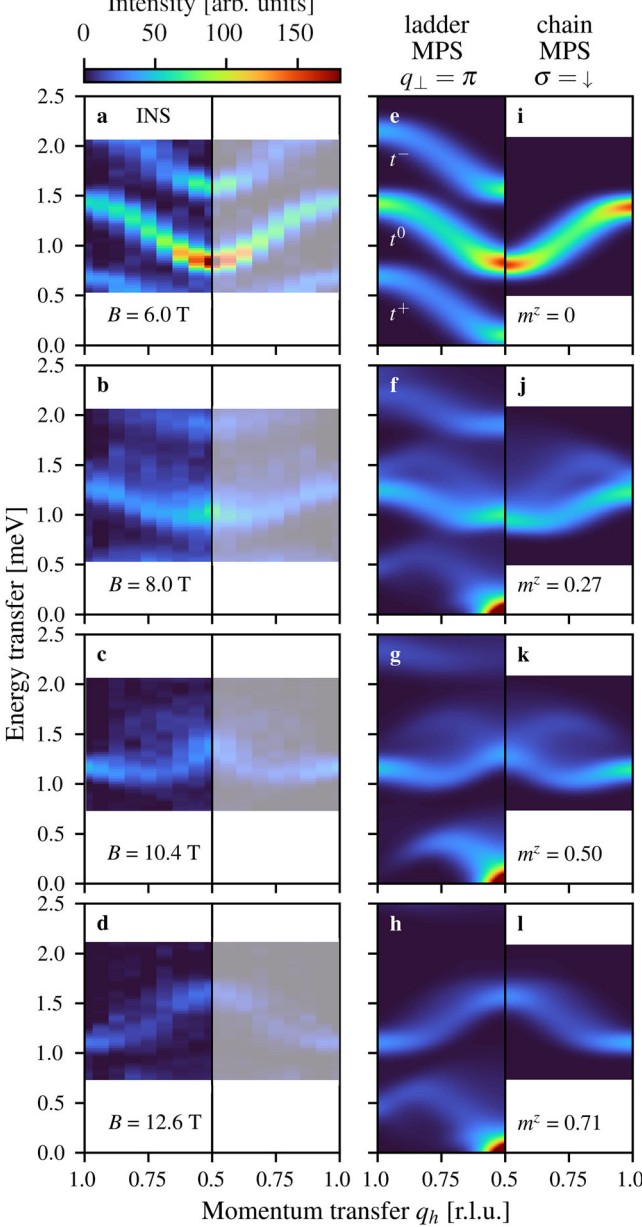

**Fig. 2 | Field-induced evolution of the magnetic excitation spectrum in BPCB.** **a**−**d** Measured neutron scattering intensities in sector $q_\perp = \pi$ at a temperature of 50 mK and applied magnetic fields of 6.0, 8.0, 10.4 and 12.6 T, where the ladder magnetizations are respectively $m^z = 0$, 0.27, 0.50 and 0.71. Spectra in the faded region were generated by symmetry. **e**−**h** Corresponding spectral functions obtained from MPS calculations performed for the magnetized spin ladder at zero temperature (l-MPS). **i**−**l**, Zero-temperature spectral functions of the $t^0$ sector calculated by MPS using the equivalent model of one hole in a modified $t$-$J$ chain (c-MPS).

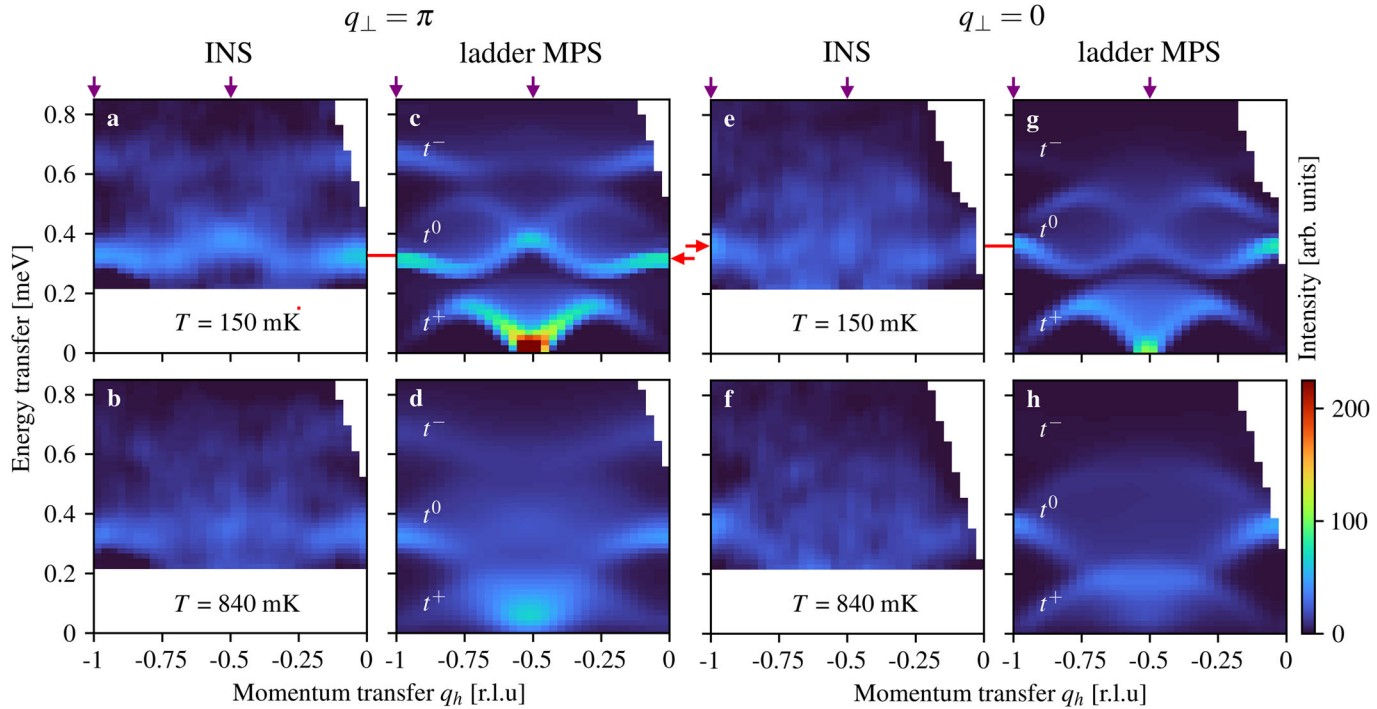

**Fig. 3 | Thermal evolution of the magnetic excitation spectrum in BPCC at $m^z = 0.5$ ($B^z = 2.876$ T). a, b** Measured neutron scattering intensities in sector $q_\perp = \pi$ at temperatures of 150 and 840 mK. **c, d** Corresponding ladder MPS calculations at 0 and 840 mK. **e, f** Measured intensities in sector $q_\perp = 0$ at temperatures of 150 and 840 mK. **g, h** Corresponding ladder MPS calculations. Vertical arrows mark $q_h$ cuts analyzed in detail and horizontal arrows draw attention to the differences between $t^0$ continua in the two $q_\perp$ sectors.

excitation, the pseudospin-1/2 system has effective Hamiltonian

$$H_{XXZ} = \sum_i \left[ J_\parallel \left( S_i^x S_{i+1}^x + S_i^y S_{i+1}^y \right) + \frac{1}{2} J_\parallel S_i^z S_{i+1}^z \right] - h \sum_i S_i^z, \quad (3)$$

with $h = b^z - J_\perp - \frac{1}{2}J_\parallel$. The $t^0$ sector is then described by a specific form of $t$-$J$ chain with an XXZ spin anisotropy and an interaction term, $J_\text{I}$, between a hole and a $\tilde{\uparrow}$ spin. A more systematic and general derivation of this model is presented in Supplementary Note 3[35]. We stress that Hamiltonians (1) and (2) provide almost identical descriptions of the $t^0$ sector, differing only by minor corrections at $\mathcal{O}[(J_\parallel/J_\perp)^2]$: we use the model of Eq. (1) in all of the ladder calculations to follow and the model of Eq. (2) in all of the $t$-$J$-chain calculations, except when explicitly stated otherwise.

Returning now to the topic of supersymmetry, despite the mystique of its predictions beyond the Standard Model, supersymmetry per se is simply an extended symmetry group that relates the operators in a Hamiltonian.[1] The supersymmetric parameter ratio of the $t$-$J$ chain studied in refs. 19–21 is obtained rather straightforwardly from the fact that the ladder model has only one energy scale for inter-rung processes, namely $J_\parallel$. Technically, the $t$-$J$ chain with a Heisenberg spin sector[19–21] has two supersymmetries, between the hole and the up-spin and between the hole and the down-spin, as depicted in Fig. 1f. With two fermions (F) and one boson (B), the system is equivalent to an FFB permutation model.[38] Quite generally, the FFB-type model with three fully symmetric species has 9 symmetry operators that make up the Lie superalgebra u(1|2), which we specify in full in Supplementary Note 3[35]. While one of these operators is the identity, one is U(1) charge conservation and three are the SU(2) spin symmetry, the four additional operations, written as the "supercharge" operators $Q_\sigma$ and $Q_\sigma^\dagger$, connect the bosonic hole to the fermionic spins and hence make these superpartners. In the ladder-derived $t$-$J$ model of Eq. (2), the two additional interaction terms combine to remove the second supersymmetry, and we will show in theory, numerics and experiment how the system

retains one exact supersymmetry with operators $Q_{\tilde{\uparrow}}$ and $Q_{\tilde{\uparrow}}^\dagger$ (Lie superalgebra u(1|1)), as depicted in Supplementary Fig. 3[35].

## Spin ladder in a magnetic field

To establish the baseline for our investigation, we look first at Fig. 2a, where a field of 6 T applied to BPCB causes Zeeman splitting of the three gapped magnon modes, which we label from bottom to top as $t^+$, $t^0$ and $t^-$. When the $t^+$ gap closes, the excitations become gapless spinons[30] and the low-energy sector of the half-magnetized ladder shows the well known des Cloizeaux-Pearson-Faddeev (dCPF)[39] continuum spectrum (turquoise shading in Fig. 1c). The field functions as an effective chemical potential for the spinons (depicted in Supplementary Fig. 4 and discussed in Supplementary Note 4[35]) and is reflected directly in the incommensurability of the spinon continuum (Fig. 2f, h). A $t^0$ excitation constitutes a third type of particle dressed by a spinon, exactly analogous to a single hole in a spin chain (Fig. 2j–l), and the supersymmetric relationship between the hole and the spinon fractions of the $t^0$ branch is the object of our current investigation. Figure 2 shows how raising the field through the gapless regime changes the shape of the $t^0$ continuum (Fig. 2b–d): while the spectral weight remains concentrated between 1.0 and 1.7 meV, the minima and maxima move systematically across the Brillouin zone as the field changes the spin polarization (and hence the Fermi wavevector of the effective spinons).

To analyze our measured spectra, we compare our BPCB results directly with the time-dependent MPS calculations[40–42] of ref. 33, and have performed analogous calculations for the parameters of BPCC. A summary is provided in the Methods section and further details of the results in Supplementary Note 5[35]. Figure 2e–h show how the emergence of deconfined spinons at the lower critical field ($B_c = 6.6$ T in BPCB) causes all three magnon branches to decompose into continua whose intensity structures evolve systematically with the field. To interpret the features of the $t^0$ continuum, we have performed additional MPS calculations for the dynamics of a single hole in different

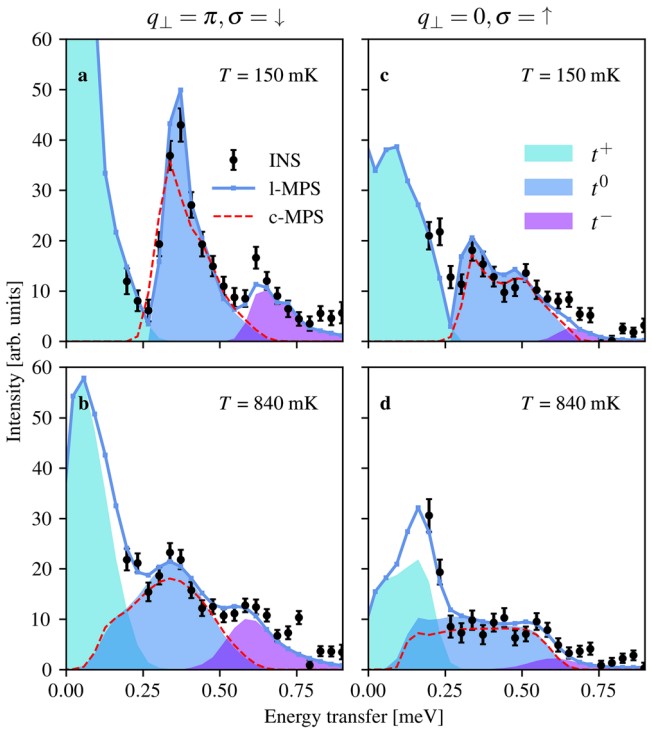

**Fig. 4 | Many-body thermal broadening.** Background-subtracted scattered intensities measured for BPCC at $m^z = 0.5$ and at low (**a**, **c**) and intermediate (**b**, **d**) temperatures, integrated over momentum transfers $q_h \in [-0.625, -0.375]$ and shown for sectors $q_\perp = \pi$ (**a**, **b**) and $q_\perp = 0$ (**c**, **d**). The error bars indicate one standard deviation. In both $q_\perp$ sectors one observes a dramatic increase in the spectral weight of the $t^0$ continuum at energies below the $T = 0$ edge. The solid blue line shows the ladder spectrum (l-MPS) and the dashed red line the spectral function of a single hole in a $t$-$J$ chain (c-MPS), both computed by zero-temperature MPS in the upper row and by finite-temperature MPS in the lower row, and with the same momentum integration, data binning and equivalent resolution as in experiment.

variants of the $t$-$J$ model (Supplementary Note 6[35]). Figure 2e–l make clear the quantitative equivalence of the two models for the $t^0$ branch.

For a comprehensive experimental analysis of the $t^0$ continuum, and hence of ladder-derived $t$-$J$ physics, we combined the capabilities of the LET spectrometer with the lower energy scales of BPCC to extend our investigation in two ways. Working exclusively at half magnetization ($m^z = 0.5$, $B^z = 2.876$ T, analogous to Fig. 2c, g and k), we measured the symmetric ($q_\perp = 0$) sector of the ladder with the same degree of accuracy as the antisymmetric ($q_\perp = \pi$) sector studied in Fig. 2. Our low-temperature results (top row of Fig. 3) show two $t^0$ continua with clearly different intensity structures and discernibly different energies, but sharing the common features that most intensity is concentrated along their lower edges and their maximal spectral weight occurs at $q_h \equiv q_\parallel/2\pi = 0$ and $-0.5$. The second extension is to repeat our measurements at finite temperatures (on the order of $J_\parallel$, bottom row of Fig. 3), and we will show why both extensions are key to revealing the effects of supersymmetry.

In the spin ladder it is known from MPS calculations[33] that the two $t^0$ continua differ, although an experimental measurement (Fig. 3) was not previously available. These differences can be regarded as a consequence of interaction effects beyond the simple picture of convolving a $t^0$ and a spinon (Fig. 1c); spinon interactions in the $t^+$ branches[43] are discussed in Supplementary Note 4[35]. Before interpreting their effects in the two $t^0$ continua, we stress a little known but fundamental property of any Heisenberg model in an applied field, that the spectral function at wavevector **q = 0** has an exact pole at the Zeeman energy[43,44]. In our BPCC measurements and MPS spectra, the

continuum response in the $q_\perp = 0$ sector sharpens dramatically at $q_h = 0$ because of this pole.

To analyze holon and spinon interaction effects in the two $t^0$ continua, we turn to the mapping of the ladder to the properties of one hole in the $t$-$J$ chain specified in Eq. (2)[33]. The key properties of the mapping are (i) the spin chain has XXZ interactions with $J_z = J_\parallel/2$; (ii) the hole hopping is given by $t = J_\parallel/2$; (iii) an interaction term, $J_1 = J_\parallel/4$, appears only between the hole and an up-spin. Property (i) gives the XXZ chain spinon excitations and leads to the "noninteracting" interpretation of the measured $t^0$ spectral function (Fig. 3) as a convolution of the elementary $t^0$ branch (the cosine observed at $B = 0$) with a single spinon of Fermi wavevector $q_\parallel = \pm \pi/2$ (blue shading in Fig. 1c)[33,45]. Property (iii) distinguishes between hole ($t^0$) states of $|s\rangle$ ($\tilde{\downarrow}$) or $|t^+\rangle$ ($\tilde{\uparrow}$) origin in the ladder, causing the two spectral functions to develop the differences we observe between the symmetric and antisymmetric sectors (shown explicitly by MPS in Supplementary Fig. 5[35]). Property (ii) establishes the two supersymmetries of the pure $t$-$J$-chain model, and in Supplementary Fig. S7[35] we illustrate how all three properties combine to preserve a single supersymmetry of the ladder-derived $t$-$J$ model, which is the duality between the $\tilde{\uparrow}$ fermion and the bosonic holon.

## Observable consequences of supersymmetry

To search for experimental observables, a first essential statement is that supersymmetry is a global symmetry: the supersymmetric operators act on the entire system, and summing over all lattice sites makes them relevant at wavevector **q = 0**. Second, the action of these operators in interchanging bosons and fermions allows exact statements about the energies in sectors of different particle number. Explicitly, if $|\psi(N_e)\rangle$ is any eigenstate of $N_e$ $\tilde{\uparrow}$ particles with energy $E$, then $Q^{(\dagger)}_{\tilde{\uparrow}}|\psi(N_e)\rangle$ is an exact eigenstate of $N_e \pm 1$ particles with energy $E \pm 2t$ (assuming $\mu = 0$)[20]. While other authors have considered sectors of arbitrary $N_e$[20,46], in the ladder-derived $t$-$J$ chain this establishes an exact correspondence between the zero- and one-hole sectors. Because the single-hole or -electron dynamical spectral function involves the matrix elements of $c^{(\dagger)}_{q\sigma}$ (which reduce to $Q^{(\dagger)}_\sigma$ at **q = 0**) between the two sectors, the fact that all states in each sector have the same energetic separation from their partners guarantees that the spectral function has an exact $\delta$-function at zero momentum and at an energy close to $\pm 2t$. Because the thermal states of both sectors still have the same supersymmetric relationship, the $\delta$-function form persists at all temperatures (up to the limit of the ladder-chain mapping, which is set by $J_\perp$). The weight of this $\delta$-function may depend on the temperature or magnetic field, but its nature and location in energy are fixed by the supersymmetry. Supersymmetry thus appears as another symmetry protecting the response, in this case from thermal broadening.

We stress that, when considering the electronic response as a convolution of spin and hole degrees of freedom in a $t$-$J$ chain, the shape of the support is not obvious from kinematics and hence the supersymmetric $\delta$-function is most intriguing. However, the fact that the model is derived from a Heisenberg Hamiltonian in a uniform magnetic field guarantees a single pole in the response function; this pole is inherited by the chain model we study, although for the most general Heisenberg ladder model (Supplementary Note 3C[35]) it can be found at different combinations of $t_\sigma/J$, the XXZ anisotropy and the strength of the interaction term.

Finding fingerprints of supersymmetry (appearing around $q = 0$) thus requires a comparison with the conventional kinematics of thermally induced many-body phenomena (appearing at generic $q$ values). In experiment, strong thermal effects on conventional spin-charge separation have been reported in the response of a 1D conductor[23]. Numerically, computing the spectrum of a 1D system at finite temperatures poses a challenge for MPS methods in controlling the spread of entanglement. We have implemented state-of-the-art MPS

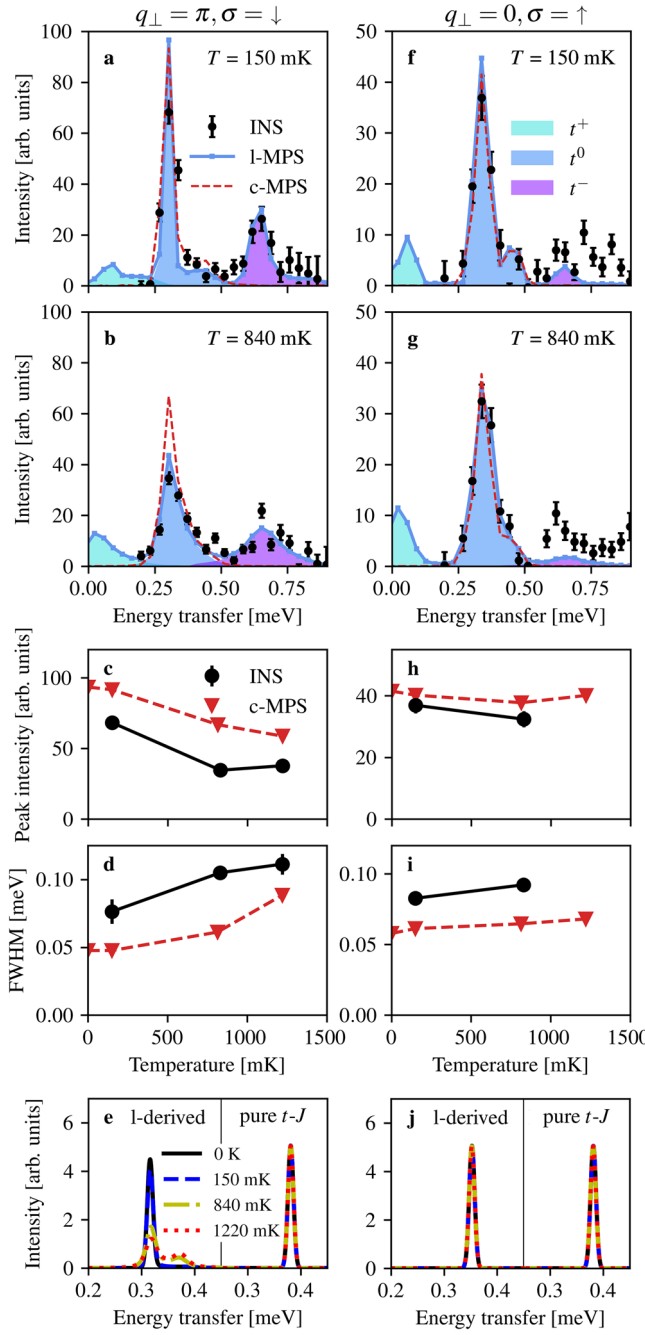

**Fig. 5 | Supersymmetric protection from thermal broadening.** Background-subtracted scattered intensities measured for BPCC at $m^z = 0.5$ at low (**a**, **f**) and intermediate (**b**, **g**) temperatures, integrated over $q_h \in [-1.125, -0.875]$ and shown for sectors $q_\perp = \pi$ (**a**, **b**) and $q_\perp = 0$ (**f**, **g**). The error bars indicate one standard deviation. The l-MPS and c-MPS lines are as in Fig. 4. **c**, **d**, **h**, **i** Heights and widths of the $t^0$ spectral peaks shown as functions of temperature and compared with c-MPS calculations. Without supersymmetry, increasing $T$ causes a reduced intensity (**c**) and broader peak (**d**), whereas in the supersymmetric sector there is no thermal evolution within the experimental and numerical resolution. **e**, **j** Spectral functions of the pure and ladder-derived $t$-$J$ chains computed at $q_h = 0$ (i.e. with no momentum integration) in both parity ($q_\perp$) sectors at $T = 0$ and for the three experimental temperatures.

techniques[42,47,48] based on density-matrix purification[49,50] to obtain accurate numerical results for both the ladder and chain models at the experimental temperatures. Because our MPS spectra offer additional insight into many-body thermal phenomena that are obscured in the INS spectra, particularly below 0.15 meV, we discuss the $t^+$ continua in

Supplementary Note 4 and the $t^0$ ($t$-$J$) continua in more detail in Supplementary Notes 5 and 6[35].

The most obvious thermal effect on the $t^0$ continua is the rapid loss of the narrow, intense low-$T$ features in both parity sectors [Fig. 3a, c, e, g] as all spectra undergo a dramatic broadening towards lower energies around $q_h = -0.5$ [Fig. 3b, d, f, h]. Some weight is also shifted upwards, creating wide distributions that become nearly uniform over their entire energy range. For a quantitative analysis, we discretize $q_h$ into three regimes by integrating over regions of width $\Delta q_h = 0.25$ centered at $q_h = -1$, $q_h = -0.75$ and $q_h = -0.5$; the most striking example of this many-body thermal broadening occurs around $q_h = -0.5$ and $q_\perp = \pi$ [Fig. 4a, b], where the high-energy half of the $t^0$ continuum changes rather little, whereas the low-energy half shows a striking shift of weight from the maximum, essentially filling in the formerly empty energy window from 0.15 to 0.3 meV. The changes in the $q_\perp = 0$ sector [Fig. 4c, d] are qualitatively the same, as in fact is the physics of the $t^-$ continuum.

Although far from the conventional thermal broadening of an integer-spin mode, this physics is generic for fractionalized excitations. At finite temperatures, the leading higher-order terms in the $t^0$ spectrum involve three spinons. The spectral response is therefore broadened vertically by all possible energies of the additional spinon pair[46], as depicted alongside MPS data in Supplementary Fig. 5[35], and the most obvious consequence at finite $T$ is to occupy additional initial states, which accounts directly for the strong spectral weight in the range $\omega = 0.15$–$0.3$ meV at $q_h = -0.5$ in Fig. 4b, d. A more detailed discussion is deferred to Supplementary Note 5, where Supplementary Fig. 6 provides the experimental data for thermal broadening in the mid-zone regime (i.e. around $q_h = -0.75$)[35].

At $q_h = 0$ (or equivalently $-1$, Fig. 5) we anticipate a special situation, whose confirmation is our central result. Here the supersymmetry should protect a $\delta$-function response at all temperatures for $q_\perp = 0$, but not for $q_\perp = \pi$. Observing this effect is complicated by the fact that the finite-$T$ $q_\perp = \pi$ peak (Fig. 5b) is already rather narrow compared to $q_\perp \neq 0$ (Fig. 4c). This is a consequence of kinematic effects also visible in Fig. 3, and is not a hallmark of any close proximity to supersymmetry (we recall that $J_l$ is on the order of $J_{\parallel}$). Nevertheless, this peak becomes half as tall and 50% broader from 150 to 840 mK (Fig. 5a–d). By contrast, the peak at $q_\perp = 0$ in Fig. 5f–g changes by less than 15% in both measures (Fig. 5h–i).

To establish a benchmark for the properties of the measured peaks, given the finite momentum integration and instrumental resolution, we use our $t$-$J$-chain calculations to obtain the dashed red lines in Fig. 5a–d and f–i. We remark here that our finite-$T$ MPS augmented by linear prediction of the time evolution connects smoothly from the spectral peak at 150 mK to $T = 0$. We stress again that our results consist of only one high-temperature dataset ($T = 840$ mK, Fig. 5g) in the supersymmetric $q_\perp = 0$ sector, with a rather low peak intensity and a broad integration window contributing an extra feature around 0.44 meV. Nevertheless, the contrasting shapes of the primary peaks in Fig. 5b, g and the agreement with MPS modeling provide quite striking evidence for the influence of supersymmetry.

In Fig. 5e, j we use our finite-$T$ MPS calculations at their highest resolution (approximately 0.025 r.l.u. in momentum and 0.012 meV in energy) to compare the supersymmetric spectral function of the ladder-derived $t$-$J$ chain with that of the pure (Heisenberg) $t$-$J$ chain of refs. 19–21 In the Heisenberg $t$-$J$ model, the peak shapes at $q_h = 0$ for both the up- and down-spin response functions are supersymmetry-protected against thermal evolution, showing no change in height or width. In the ladder-derived model, the supersymmetry between the charge and the down-spins is broken, resulting in a pronounced temperature-dependence (Fig. 5a–e), in strong contrast to the persistent $\delta$-function nature of the up-spin response (Fig. 5f–j). We remark here that nothing close to a $\delta$-function response at $q_h = 0$ has been obtained in studies of cuprate $t$-$J$ chains ($t = 3J$).[24-26]

## Discussion

The fractionalization of collective excitations into underlying fermionic and/or bosonic components can nowadays be called commonplace in models of correlated condensed matter. Still it is rare that such models are considered from the standpoint of supersymmetry, and although to date no supersymmetric model proposed in the condensed-matter literature has been close to an experimental realization, this situation is advancing in quantum optics as realized in synthetic quantum systems[12–15]. Here we have shown how the two-leg quantum spin ladder in an applied magnetic field can be used not only as a "quantum simulator" for the paradigm problem of a hole propagating in a highly correlated spin background, but that the associated $t$-$J$ model is supersymmetric. Although we do not obtain any of the spectacular consequences desired of supersymmetry beyond the Standard Model – hitherto unknown superpartner particles at exotic energy scales reflecting broken supersymmetry – we do find (i) a clear realization of an expanded supersymmetric group containing dual bosonic and fermionic superpartners, (ii) demonstrable consequences in the spectra of the zero- and one-boson sectors and (iii) supersymmetry-enforced kinematic constraints that become particularly evident in the presence of thermal fluctuations.

Away from the supersymmetric point, the doped $t$-$J$ model still captures one of the most complex problems in condensed matter, specifically of a fractionalized "impurity" particle in a fractional spin environment, and with the quantum spin ladder this can be studied away from the supersymmetric wavevector ($\mathbf{q} = \mathbf{0}$). From an experimental standpoint, the equivalence of the ladder to a $t$-$J$ model allows the energy and momentum resolution of modern INS spectrometers to be applied to probe fractional impurity dynamics at levels of accuracy, homogeneity, system size and low effective temperatures not yet accessible to other experimental platforms such as ultracold trapped atoms or ions. The two control parameters of the applied field, which modulates the spatial response of the environment, and the temperature, which alters the availability of states at different energies, allowed us to observe coherent quantum many-body phenomena where thermal fluctuations drive dramatic qualititative changes to the spectrum.

While the ladder-to-chain mapping does not produce precisely the Heisenberg $t$-$J$ chain, the interaction term provides an additional "handle" that selects quasiparticle spin states preserving or breaking supersymmetry. It is crucial that this interaction is constrained to preserve one supersymmetry, and hence it cannot alter the qualitative nature of the spectra by forming bound states that split off from the continua. Varying the ladder parameters provides the insight that there exists a large family of ladder-derived $t$-$J$-chain models, up to and including those with strongly asymmetric spinon species akin to a Falicov-Kimball model, whose Heisenberg-ladder origin nevertheless ensures one exact pole in the spectral function and one supersymmetry, even if the model is no longer integrable (or has very well hidden integrability).

We note again that the family of ladder-derived $t$-$J$-chain models lies far from the regime $t \gg J$ describing insulating spin chains with true electronic degrees of freedom. In the ladder-derived models, it is clear from our analysis that a spinon-holon description remains robust, but the interpretation of the spectral features is not as straightforward as in the large-$t$ regime, where the lower continuum edge was associated directly with the response of a single spinon[25,26]. In this regime, the finite-temperature spectral function[51] reflects the physics of the spin-incoherent Luttinger liquid[52,53], where the charge and spin degrees of freedom decohere at quite different temperatures (characterized respectively by $t$ and $J$). However, in the supersymmetric $t$-$J$ model, the two sectors have the same energy scale and we obtain the physics of the $\mathbf{q} = \mathbf{0}$ pole that persists up to temperatures set by the energy scale for coherence of the ladder rung states, which is $J_\perp$.

To summarize, the study of partially polarized two-leg spin-1/2 ladders opens the door to deeper insight into the physics of fractional and supersymmetric quantum particles. The massless continuum can be used to probe spinons with field-controlled incommensurability and the massive continuum to probe supersymmetric holon-spinon fractionalization dynamics, both with interaction control, sector separation and, unless supersymmetry-forbidden, many-body thermal renormalization effects extending to energies far in excess of $T$. Where supersymmetry does forbid thermal broadening, it emerges as another candidate for the protection of quantum mechanical information from decoherence. Supersymmetry therefore continues to be a valuable organizing principle throughout physics: its application to the fermionic and bosonic fractions found in correlated condensed matter is in its infancy, and here we set in motion its experimental investigation.

## Methods

### Materials

Large and high-quality single crystals of both BPCB and BPCC were grown by a method of slow solvent evaporation[54]. Further details are provided in Supplementary Note 1[35]. The two materials are isostructural, with monoclinic space group P2$_1$/c, and their low-temperature structures have been determined by neutron diffraction (Supplementary Note 1)[32,34]. The crystal symmetry mandates two types of ladder with identical interaction parameters but two different orientations. The ladder legs are aligned along the crystallographic $a$ axis and the rung alignment vectors are $\mathbf{r}_\pm = (0.3910, \pm 0.1625, 0.4810)$ for BPCB and $\mathbf{r}_\pm = (0.3822, \pm 0.1730, 0.4866)$ for BPCC, expressed in fractions of the unit cell. For both INS experiments, several crystals were co-aligned within 1˚, aided by neutron diffraction on the MORPHEUS instrument at the Swiss Spallation Neutron Source, SINQ, at the Paul Scherrer Institute (PSI), and glued onto a gold-coated aluminum holder. The mount for the BPCB experiment consisted of six co-aligned crystals with a total mass of 1.4 g. The mount of seven BPCC crystals with total mass 3.535 g is shown in Supplementary Fig. 1[35].

### Inelastic neutron scattering measurements

The BPCB experiment was performed at the triple-axis spectrometer ThALES at the Institut Laue-Langevin (ILL, Grenoble, France), which has a 15 T vertical-field magnet. The final neutron momentum was set to 1.3 Å$^{-1}$ by a PG(002) monochromator with a Be filter, which allowed a resolution of 0.16 meV in energy and 0.14 r.l.u. in momentum along $a^*$, and the measurement temperature was 50 mK. Further details are provided in Supplementary Note 2[35].

The BPCC experiment was conducted at the time-of-flight spectrometer LET[37] at the ISIS Neutron and Muon Source at the Rutherford Appleton Laboratory (Chilton, United Kingdom), which has a 9T vertical magnet. The incoming neutron energy was set to 2.5 meV and the sample was rotated in 1° steps, spanning a total of 103°, around its $c$ axis, which was set parallel to the field. The energy resolution at $q_h = 0$ varied from 37 $\mu$eV at the elastic line to 28 $\mu$eV at an upper energy transfer of 0.9 meV, and at $q_h = -1$ from 56 $\mu$eV to 34 $\mu$eV. For the purposes of analyzing the $\delta$-function peaks in the scattered intensity, we note that the energy resolution at 0.3 meV for $q_h = -1$ was 52 $\mu$eV in both $q_\perp$ sectors. The momentum resolution was approximately 0.05 r.l.u. along $a^*$. Scattering intensities were corrected using MANTID,[55] binned into $S(\mathbf{Q}, \omega)$ datasets[32] and integrated using HORACE[56]. Our raw scattering intensity data are shown in Supplementary Fig. 2 as part of the description of our data analysis and background-subtraction methodology in Supplementary Note 2[35].

### Ladder and chain MPS

To calculate the spectral function of the two-leg ladder, and of different $t$-$J$-chain models, both at zero and at finite temperature, we use MPS methods in real and imaginary time[40,41]. We compute the two-time

real-space correlation function

$$C(x_0, x_1; 0, t) = \left\langle B^{\alpha}_{x_1}(t) B^{\gamma}_{x_0}(0) \right\rangle_{\beta}$$

at zero temperature, or at inverse temperature $\beta$, where $B^{\alpha}$ represents either the spin operators in the ladder (with $\alpha = \pm, z$) or the fermionic operator in the $t$-$J$ model. In space, the correlation function is centered at rung $x_0 = L/2$ of the ladder or site $x_0 = L/2$ of the chain, where $L$ is the system length. In time, we performed real- and imaginary-time evolution using the time-evolving block decimation (TEBD) method in our ladder calculations and the tMPS method for the chain, with second- or fourth-order Trotter-Suzuki decomposition and purification of the density matrix at finite temperature[47]. The codes for our ladder calculations were developed in-house and those for the $t$-$J$ chain were developed based on the ITensor library[57].

The MPS spectral functions at different temperatures and applied magnetic fields were obtained from $C(x_0, x_1; 0, t)$ by a discrete Fourier transformation to frequency-momentum space. This process conventionally uses a Gaussian filter function, $M(x, t) = e^{-(x/x_w)^2} e^{-(t/t_w)^2}$, where the filter widths in space and time, $x_w$ and $t_w$, can be used to remove finite-size effects[48]. We used our MPS spectral functions for two purposes. To identify the qualitative and quantitative properties of our ladder and chain spectra, we maximized the system length and evolution time, the latter by implementing linear prediction[58], to optimize the resolution in momentum and frequency. For comparison with experiment, we included the ladder orientations of the real material, weighted the numerical structure factors by the **Q**-dependent scattering terms, integrated over **Q**-space in the same way as our treatment of the experimental data and selected appropriate filter parameters. Explicit technical details for achieving each task, meaning system sizes, time steps, filter widths and truncation errors, are listed in Supplementary Notes 5 and 6[35].

## Data availability

The raw INS datasets from this study are available at https://doi.org/10.5286/ISIS.E.RB1410142. All l-MPS, c-MPS and processed INS data shown in the figures and otherwise supporting the results of the manuscript may be found at https://zenodo.org/records/14989856.

## Code availability

Neutron scattering data were processed using the open-source software MANTID[55] and HORACE[56] as detailed in the Methods section. The codes used for MPS calculations are available from the corresponding author on request.

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

## Acknowledgements

We thank T. Barthel, C. Berthod, F. Essler, V. Gorbenko, T. Guidi, N. A. Kamar, D. McMorrow, M. Mena, M. Spira and S. Takayoshi for helpful discussions. This work was supported in part by the European Community Seventh Framework Programme (FP7/2007-2013) under Grant Agreement No. 290605 (PSI-FELLOW/COFUND) (B.W.), by the Swiss National Science Foundation (SNF) under Grants No. 200020-132877 (Ch. R. and K.K.), No. 200020-188687 (T.G.) and No. 200020-219400 (T.G.) and by the Deutsche Forschungsgemeinschaft (DFG, German Research Foundation) under Grants No. 277625399-TRR185 (B3) (C.K.), No. 511713970-CRC1639 NuMeriQS (C.K.) and No. 277146847-CRC1238 (C05) (C.K.), as well as through the Cluster of Excellence Matter and Light for Quantum Computing (ML4Q) EXC 2004/1-390534769 (F.L. and C.K.). We acknowledge the provision of neutron beam-time at the ILL within proposal 4-03-1598, at ISIS within experiment number RB1410142 and at the Swiss Spallation Neutron Source, SINQ, and we thank the technical and scientific staff for support at all three facilities. Calculations were performed at the University of Geneva on the Baobab and Mafalda clusters and at the University of Bonn on the BAF cluster.

## Author contributions

B.W., Ch.R., A.L. and T.G. designed the study. D.B. and K.K. grew the single crystals. B.W., S.W., B.T., M.B., R.B and Ch.R. performed the INS experiments and analyzed the data. F.L., N.K., P.B. and B.W. performed MPS calculations with input from T.G., C.K., B.N. and A.L. The manuscript was written by B.W., F.L., C.K., A.L. and B.N. with input from all the authors.

## Competing interests

The authors declare no competing interests.
