## [Transparent Peer Review file · Nature Communications]

Fingerprints of supersymmetric spin and charge dynamics observed by inelastic neutron scattering

Corresponding Author: Dr Bjoern Wehinger

Version 0:

Reviewer comments:

Reviewer #1

(Remarks to the Author)

The authors present a comprehensive experimental and numerical study of the spectral properties of the quantum spin ladder compounds BPCB and BPCC under a magnetic field. In this regime, it has been demonstrated that these strong rung ladder systems can be effectively mapped onto an equivalent t-J chain model where triplet excitations play the role of the holes. The mapping is well established and corroborated by numerical simulations. In this manuscript, the authors bring up the fact that the effective t-J chain model falls in a parameter regime that coincided with the supersymmetric exactly soluble (integrable) point $J=2t$. Despite the fact that it's an XXZ-type model, the consequences are observable and are manifested particularly by a robust pole at the zone center. The comparison with DMRG calculations is remarkable, and establishes these materials as bona fide quantum simulators. The study of supersymmetry in a toy model like this can allow experimentalist to peek into the properties of supersymmetric quasi-particles. The paper is sound, well written, and of high quality and broad interest, and potentially highly impactful. I recommend it for publication as is.

Reviewer #2

(Remarks to the Author)

The authors present experimental inelastic neutron scattering results on two spin-ladder compounds $(\text{C}_5\text{D}_{12}\text{N})_2\text{CuBr}_4$ and $(\text{C}_5\text{D}_{12}\text{N})_2\text{CuCl}_4$ at variable fields and temperatures. The behavior of the materials are considered within a t-J model. As a central result a consideration of the interaction of spin (fermion) and charge (boson) is undertaken to probe the supersymmetric nature of these materials. Such experimental considerations of these interactions are rare and this approach is likely to be of interest to readers, with the potential to open up new research directions. One concern is that the model chosen is one of several that could describe the data and as such is not unambiguous evidence for supersymmetry. However, the analysis appears robust and the conclusions are similarly carefully considered.

Comments for consideration:

- It is not clear why 2 different materials were chosen. This have similar $J_{\text{perp}}/J_{\text{parallel}}$ ratios, but different absolute values (~3 times different J_{perp}).
- The data on the BPCB is markedly better than that on BPCC due to the different instruments used. To this point, the BPCC data (lower quality) is used to show the main results of the manuscript, Fig 3, Fig 4 and Fig 5. The data are "background" subtracted. Was this background subtraction from a measurement of the background or from a calculation. The raw unsubtracted data should be shown in the supplemental information.
- The authors must clearly state the energy resolution of the measurements at LET. In particular at energies of the to scattering? This is required to interpret the result in Fig 5 in particular. The resolution of THALES is given.

Reviewer #3

(Remarks to the Author)

See attachment.

Version 1:

Reviewer comments:

Reviewer #2

(Remarks to the Author)

Having reviewed the responses and the associated revisions the manuscript is now suitable for publication.

Reviewer #3

(Remarks to the Author)

In the revised version of their manuscript the authors have addressed the Reviewers' comments and I can recommend publication. Below is a list of minor suggestions which I leave to the authors' consideration.

line 54: Another proposal to implement exact lattice SUSY is [R1].

line 192: The statement "we will use this model" is ambiguous – do the authors refer to the Hamiltonian (1), (2) or H_{XXZ} ?

[R1] J. Minar et al., Phys. Rev. Lett. 128, 050504 (2022)

Response to the Report of Reviewer #1

We would like to thank the reviewer for such a concise and accurate summary of our work, highlighting directly its value in the context of quantum simulation and supersymmetry. We are especially grateful for the rarely-given accolade that it may be published without change.

Response to the Report of Reviewer #2

We thank reviewer for a detailed reading of our contribution and for posing a number of questions that help us to clarify the presentation of both our experimental and theoretical results.

On the theory side, the referee asks one question which seems to indicate a misunderstanding: the statement “the model chosen is one of several that could describe the data and as such is not unambiguous evidence for supersymmetry” is not correct. There is only one model to describe the data for the scattering response in the t^0 sector. It is equivalently stated as a Heisenberg spin model on a ladder geometry or as a t - J model on a chain geometry with 1 hole. This t - J model has the very specific form spelled out in Ref. [29] and repeated in our manuscript. We reiterate that the two models are equivalent and there is no ambiguity in which model to use. In our manuscript we did not discuss any other models, apart from the t - J model with a purely Heisenberg spin sector for illustrative comparison of qualitative features. We have reviewed and reworded the revised manuscript very carefully to avoid any possibility that the reader could become confused over the model, and thank the referee for alerting us to this danger. Finally, we note that there are definite reasons why our work might be taken as failing to provide unambiguous evidence for supersymmetry, a topic we take up again in our response to Reviewer #3, but uncertainty over the correct model to apply is not one of them.

On the experimental side, the referee notes correctly that we study two different ladder materials. This was a deliberate choice to stress the universality of our considerations. On the pragmatic side, the factor-3 difference in energy scales noted by the referee translates directly to a factor-3 difference in applied fields and temperatures at which the phenomena we study are manifest. The high magnetic field available on ThALES made it possible to study the full range of ladder magnetizations in both materials. The lower energy scale of BPCB made it a better choice for precision studies at finite temperatures without introducing the possibility of additional physical processes and the higher background contributions we would have obtained at the equivalent temperature in BPCB.

The referee comments that our BPCB data, taken on the LET spectrometer, are of lower quality than our BPCB data (taken at ThALES). We cannot agree with this characterization. ThALES is a triple-axis spectrometer and as such has very limited coverage in Q . LET is a time-of-flight instrument and therefore has excellent Q coverage, vastly increasing the overall data quantity. This in turn makes it possible to conduct much more detailed data analysis, particularly in comparing the symmetric and antisymmetric scattering sectors. LET also has higher energy resolution (below), which was of special value in studying the very sharp (δ -function) peak associated with supersymmetry. As a result of these attributes, our LET data was also of significantly more value for detailed comparison with our MPS calculations, which also have high energy- and Q -resolution.

In connection with the previous point, the referee asks about our background subtraction. This was conducted on the basis of measurements made in different parts of Q . Following the referee’s request, we have added some raw (unsubtracted) data to the revised manuscript, in the new Fig. S2 (Sec. S2), where we also explain in a new text paragraph how the background subtraction was performed.

The referee also asks for more explicit statements of the energy and momentum resolution at LET, which we are happy to provide in the Methods section and in more detail in the revised Sec. S2. We trust that these do now allow a full interpretation of our results in Figs. 4 and 5, and thank the referee once again for helping to make our results more accessible to the reader in this way.

Response to the Report of Reviewer #3

We would like to thank the reviewer for such a detailed commentary on our study and for agreeing with us that our intriguing results merit publication in a forum such as Nature Communications. We are also grateful to the referee for drawing our attention to so many possible sources of ambiguity in our presentation, which we detail and attempt to remedy in the following.

The referee finds the arrangement of our presentation in the main text and SI “cumbersome.” In the original version this presentation was governed by the desire not to overburden the reader with a large number of definitions and discussions before we could begin to discuss the physics. Although this approach appears to have worked for Reviewers #1 and #2, we are happy to heed Reviewer #3’s advice that it may not work for most readers. We have rearranged the first section of the manuscript after the introduction in an effort to improve this situation. Here we introduce first the ladder Heisenberg Hamiltonian and then the effective t - J -chain model for the t^0 sector, and then contrast supersymmetry in the latter model with that in the Heisenberg t - J chain. In doing so, we restrict the supercharge operators in the main text and figures to Q and Q^+ , reserving the $J_1 \dots J_9$ notation for the SI to avoid overcomplication, and we trust that the referee can understand our logic for this.

The referee finds it important to stress that the supersymmetry we study is one “surviving” in the ladder-derived t - J -chain model. We do not share the perspective that this “survival” is somehow accidental or luck, and thus we do not feel the need to overemphasise it. As we have tried to make clear, the presence of one supersymmetry in the t - J -chain model is preprogrammed by the fact that the starting ladder model is a Heisenberg spin system in an applied field. We have reviewed the relevant discussions in the manuscript in order to articulate this point appropriately.

Although the referee introduces the notation $(0, \uparrow)$ and (π, \downarrow) , we stress again that $0 = \uparrow$ and $\pi = \downarrow$. These are the corresponding sectors of two models that are exactly equivalent at the 0- and 1-hole level (i.e. there is no $(0, \downarrow)$ sector).

Although the referee complains about confusing notation, she or he understands our meaning exactly. We take this as a basis on which to retain our notation l-MPS and c-MPS: all of our numerical calculations were performed on (i) a ladder Heisenberg model and (ii) a t - J -chain model. We have worked harder in the revised manuscript to stress that “c-MPS” refers to the ladder-derived t - J -chain model at all times, other than the limited cases (qualitatively in Fig. 1f and quantitatively in Figs. 5e, j and S7) where we contrast with a Heisenberg t - J -chain model (and an XXZ t - J -chain model in Fig. S7) to illustrate the number of supersymmetries (0, 1 or 2).

The referee queries the ratio of peak heights between 150 mK and 840 mK in Figs. 5a-b and on the left side of Fig. 5e. The referee is correct that Fig. 5e is shown only for $q_h = 0$, meaning without the momentum integration included in Figs. 5a-b. The referee’s intuition is also correct: in Fig. 5e we did not include the scattering cross section (from the full expression in Eq. (S9)) because this panel was intended only as a qualitative illustration of non-supersymmetric thermal suppression, and not as a cross-reference with our fully quantitative results. We apologise for this confusion.

We confirm that, unless otherwise stated, we use c-MPS to refer to the ladder-derived t - J -chain model that describes the t^0 sector of the Heisenberg spin ladder, and this is why the comparison between experiment and numerics in Figs. 4-5 is quantitatively accurate (the one-hole sector of this t - J model is exactly equivalent to the t^0 sector of the ladder Heisenberg spin model).

We confirm that there is only 1 finite-temperature dataset, $T = 840$ mK, in the $q_{\perp} = 0$ sector. We stated this explicitly in our text. The reason for this situation was the limited beamtime available for our BPPC experiments, which already totalled 25 days. In this sequence of experiments, we looked first in the sector with higher intensity ($q_{\perp} = \pi$) to establish the many-body thermal broadening and other physics less challenging than supersymmetry. This is also the reason that we kept the supersymmetry result until last in our presentation sequence. With respect to the comment of Reviewer #2, we agree in full that our results cannot be called unambiguous evidence for supersymmetry, but not for reasons of model uncertainty, only of data quantity. We are also scientists and have nothing to hide here: we feel that it is still worth writing this manuscript with the experimental data in its current form, supported by numerical calculations, to draw our indicative result to the attention of the community. It will be a long time until our INS experiment can be repeated, and our hope is to stimulate different studies of supersymmetry (for example using different materials or different experimental methods).

Concerning the effects of temperature, in Sec. S6 of the SI we commented that the upper temperature limit for our considerations is actually the effective charge gap of the chain, which is set by the rung interaction, J_{\perp} , of the strongly coupled ladder. Below this energy scale, we still expect to observe the physics of the t - J chain dictated by the rung singlet and triplet states. Following the remark of the referee, we have included a comment to this effect in the penultimate paragraph of the main text.

The referee complains of missing a “clear analytical derivation/proof” of the statement that the pole in the spectral function of the t - J -chain model is the consequence of its supersymmetry. Purely in the context of the t - J -chain model [in the Heisenberg spin ladder, the pole appearing at $q = 0$ and the Larmor energy (i.e. set by the field) is a consequence of the SU(2) symmetry], the supersymmetry enforces exact relations between the spectra of the sectors with different numbers of excited holes. Here we focus only on the zero- and one-hole sectors, which are the only ones required to compute a single-particle excitation spectrum. The supersymmetry guarantees that every state in the $q = 0$ manifold of the zero-hole sector has a partner in the one-hole sector at exactly the same energetic separation (which is close to $2t$ in the ladder-derived t - J chain, while in the Heisenberg variant it is precisely $2t$). This situation guarantees that the spectral function $\langle c^{\dagger}c \rangle$, which becomes equivalent to $\langle Q^{\dagger}Q \rangle$ at $q = 0$, has a δ -function pole at this energy, as no other energies are represented. Furthermore, it remains true at finite temperatures, where the thermal states of both sectors still have the same supersymmetric relationship in energy. In the revised manuscript we have included a detailed explanation of this reasoning and hope that this satisfies the referee’s request.

Turning to the referee’s brief remarks:

The ladder rung singlet is not present in Fig. 1b, which shows the singlet-triplet response.

The referee is correct that the singlet and triplet information can usefully be included in Fig. 1d.

We have followed the referee’s advice and introduced a $\tilde{}$ notation for the effective spins of the t - J model and have made clear that these are the red spins depicted in Figs. 1 and S1.

We find that the average magnetization was introduced in the main text, particularly in Fig. 2, and have made this more explicit in the revised manuscript, while also unifying on the notation m^z .

The existence of Falicov-Kimball models (meaning based on two species of particle with very different hopping amplitudes) that have one supersymmetry can be read directly from Eq. (S7) with large J_x . It is an original observation of the present work and cannot be referenced to any other authors; we include it to indicate some added value of our present results beyond the confines of the $J_x = 0$ model we study numerically, and trust that the reviewer can accept this approach. Indeed the one supersymmetry has the same origin as the one we expose in detail, namely the Heisenberg ladder origin, and we did not see a need to repeat this point again.

In the revised manuscript we provide the exact values of B_c and B_s for both BPCB and BPCC, and in the revised Fig. S3a we show more explicitly how the variation of m^2 as a function of the applied field translates to the incommensurate field-induced response measured and computed in Fig. 2.

The referee asks about the connection between the former Sec. S1A and the supersymmetric t - J and ladder-derived t - J models. Certainly we intended Sec. S1A as the most basic introduction to supersymmetry, whose primary aim was to illustrate the presence of paired (superpartner) bosonic and fermionic states. For this purpose we believe it to serve as a suitable introduction. We agree in full with the referee's statement that supersymmetry in t - J models, which are extended systems, has a structure different from, and rather more complex than, the basic $\{Q, Q^+\}$ form. We have added a sentence to this effect in what is now Sec. S3, and thank the referee for helping us reduce possible confusion by ensuring that the reader does not anticipate a direct connection.

We thank the referee for stressing to us the need to be more explicit in differentiating supersymmetric systems with $E = 0$ states from those without: in a condensed-matter system with a variable (and in our case field-controlled) chemical potential, we did not dwell on the possibility of $E = 0$ states and their relation to a supersymmetric ground state. In the revised manuscript we have reordered Sec. S1A (now Sec. S3A) to state more clearly which aspects of the conventional introduction we follow and which we can neglect for our model and purposes, and we trust that the referee can now accept this as a suitable introduction for a non-specialist reader.

Finally, we are grateful to the referee for drawing our attention to the work of Cai *et al.*, which is a valuable complement to our work: we perform spectroscopy while these authors perform quantum state tomography; we work with an extended system while these authors consider a single trapped ion; we focus on what is conventionally called "broken supersymmetry" (terminology we do not favour, given that the Hamiltonian is supersymmetric, but meaning no $E = 0$ states) while these authors focus on contrasting the cases with and without $E = 0$ states (previous point). We have commented on the work of these authors, and others proposing to study supersymmetric models achievable with cold atoms, at 2 points in the revised manuscript.

We would like to thank the referee very much once again for such a systematic appraisal of our study and for so much assistance in explaining our results to the reader more clearly. We trust that the referee will be able to agree that the revised manuscript brings our key messages home in a more intuitive and less potentially confusing manner, and as a result that our work may now be appropriate for publication in Nature Communications.

We are very grateful to both referees for their positive assessment of our revised manuscript. In response to the remaining comments of Reviewer #3, in line 54 (and also in line 461) we have added the work of Minar *et al.* (now Ref. [14]) to the manuscript. In line 192 we have made a very explicit statement of where we use the Hamiltonians that we stated in the previous line are equivalent.

In their manuscript “Fingerprints of supersymmetric spin and charge dynamics observed by inelastic neutron scattering” Wehinger and colleagues present a study of a neutron scattering experiment on two compounds – BPCB and BPCP – and interpret the scattering data in terms of the underlying theoretical models of the material, the anisotropic Heisenberg spin ladder and the ladder-derived effective t-J model. In particular, they interpret the observed spectral functions in specific momentum and spin sectors as a consequence of the underlying supersymmetry. Building on their previous works (including Refs. 25-29) the authors present a complex theory-experiment collaboration with intriguing results. In my opinion, the article deserves publication in some form provided the following concerns are cleared.

The starting point of the theoretical considerations is the anisotropic Heisenberg chain. In the limit of large J_{\perp} , one obtains an effective ladder-derived t-J model (tJl). It differs from the usual tJ model (tJ) in that while the latter possesses a set of 9 supersymmetry generators (one being trivial), the former inherits only two of them. The consequences of these two (J_4 and J_5 discussed in S1.B of the Supplementary Information) are then probed in the experiment.

While this information is provided by the authors, I find the way of presenting the results quite cumbersome, admixing theoretical and experimental parts and deferring some of the important information to the Supplementary Information (SI), cf. the remarks below.

I think that it would be very helpful to the reader to streamline and compactify the information. As the main claim of the paper is observing fingerprints of supersymmetry, it is essential to emphasize that it is the supersymmetry “surviving” in the tJl model. For instance one could list the tJl model together with the Hamiltonian (1) and specify the relevant supercharges $Q_{\alpha} = \sum_i J_{\alpha}^i$, $\alpha=4,5$, and their properties, $[H_{tJl}, Q_{\alpha}] = 0$, in the main text.

Next, the main claim of the paper is based on the data presented in Fig. 5. The observed persistence in the spectral function in the t_0 branch and in the $(q_{\perp}, \sigma)=(0, \text{up})$ and $q_{\parallel}=0$ (or equivalently $q_{\parallel} = \pm 1$) sector as the temperature is increased is attributed to the underlying supersymmetry (Fig. 5f,g). This is then contrasted with the decrease of the analogous peak in the (π, down) sector, Fig. 5a,b. This difference is then corroborated with modelling the tJ model and the ladder-derived tJ model in Fig. 5e,j. The dependence of the height as well as the full-width at half maximum of the peaks is further studied in panels c,d,h,i. Furthermore, these results are contrasted with the ones corresponding to $q_{\parallel} \neq 0$ sector, where no SUSY protection is expected and which are presented in Fig. 4.

My main concern is the following. First, I find the nomenclature rather confusing – the caption of Fig. 4 states that the chain MPS (c-MPS) simulations depicted as the red dashed lines in Figs. 4 and 5 correspond to the “t-J chain”. Presumably the authors refer to the ladder-derived tJ model, i.e. the one which is denoted as “tJl” in Eq. (S6) and “l-derived” in Fig. 5e,j? Similarly, the “Heisenberg” in Fig. 5e,j refers to the standard tJ model, not the Heisenberg Hamiltonian Eq. (1), which is the starting point. This should be clarified and it would be helpful if a unified notation is used throughout the manuscript.

If c-MPS indeed corresponds to the ladder-derived t-J model, then it is not clear why the peak (red-dashed line in Fig. 5b) is higher than the one presented in panel e (yellow line) at $T=840$ mK. I understand that the data in b are obtained by integrating over a range of q_{\parallel} , while the theory curves in e are for a single q_{\parallel} , but naively I would expect even more dramatic decrease of the peak when integrated over a range of q_{\parallel} . In the opposite case, if c-MPS refers to the tJ model (not the ladder derived), then it is not clear why it captures quantitatively well the data presented in Figs. 4,5.

Next, and most importantly, the abstract claims that the “persistence (of the peak in the spectral function) at all temperatures constitutes an observable consequence of supersymmetry”. While the data in Fig. 5b,g correspond qualitatively to the expected trend (i.e. SUSY protection in the $(q_{\perp}, \sigma) = (0, \text{up})$ sector), they represent a **single set** of such data at high temperatures and it is thus difficult to draw an unambiguous conclusion about the origin of the observed effects. In particular, I wonder why there is a missing data point at $T=1220$ mK for the $(0, \text{up})$ sector, Fig. 5h,i, while there is one for the (π, down) one, Fig. 5c,d? The trend of the decreasing peak amplitude and increasing width appears in both sectors, although in the $(0, \text{up})$ one it is less pronounced. Since the effective SUSY of the tJl model holds only for the low energy sector (SUSY between the zero and one-hole sectors of the tJl model), I would expect the SUSY-protected peak to decrease for temperatures $T > J_{\text{parallel}}$, the characteristic scale of the tJl model? I haven’t found the discussion of this issue in the manuscript.

Further question concerns the claim that the pole in the up-up correlation function resulting in the delta-peak in Fig. 5f,g,j is the consequence of supersymmetry. While it is manifest from the numerical simulations, also in the section S6 of the Supplementary Information, that the up-up correlation function in the $(0, \text{up})$ sector is insensitive to the temperature, and while the authors provide a qualitative discussion of these results invoking supersymmetry (e.g. the discussion at the beginning of page 3 of the SI), I am missing a clear analytical derivation/proof of this statement. For example are the theoretical expressions entering the correlator computations invariant under the action of the supercharges?

Next, I have the following remarks.

- For completeness, it might be helpful to also include the singlet branch in Fig. 1b as done e.g. in Ref. 29.
- While a detail, it might be useful to write the $|s\rangle$ and $|t_{0,+,-}\rangle$ states explicitly, $|s\rangle = 1/\sqrt{2}(|\text{down up}\rangle - |\text{up down}\rangle)$ as their pictorial representation in Fig. 1d might seem ambiguous.
- In line 268,269 the effective spins describing the rung configurations, such as $|\text{down}\rangle = |s\rangle$ etc. are introduced. This notation can easily cause confusion as the original Heisenberg Hamiltonian Eq. (1) and the spins of the ladder are also denoted as $|\text{up}\rangle, |\text{down}\rangle$. It might be helpful to emphasize that the new $|\text{up}, \text{down}\rangle$ are effective spins, by denoting them e.g. by a tilde as done for instance in Ref. 29.
- It would be also helpful to define the average magnetization m_z , introduced only in the SI, in the main text (and unify m_z vs. m^z).
- Around line 461 the authors discuss the existence of a family of ladder derived t-J models akin to Falicov-Kimball model mentioning that they possess “one supersymmetry”. It is however unclear which supersymmetry is being discussed nor what its existence implies – presumably the existence of the pole in the spectral function, but this is not justified neither a reference is given at this point, cf. my remark above.
- The value of the critical field B_c is provided (for the BPCB compound). It might be useful to provide also the value of B_s .
- In Sec. S1.A of the SI the SUSY quantum mechanics is introduced including the “exact SUSY” models, which can be written as $H = \{Q, Q^{\dagger}\}$ with nilpotent charges and some of the consequences,

such as the existence of the doublets in the spectrum, is briefly discussed. It is however not clear how this helps the subsequent discussion of the SUSY in tJ models, including the ladder-derived ones, where the supersymmetry is of different structure (no global nilpotent supercharges are provided). The presence of Sec. S1.A thus raises the question of what is its relation to the SUSY in the tJ models, which I think is not clearly explained.

- It is worth to note some recent developments regarding exploration of SUSY manifestation in trapped ions [M.-L. Cai et al. Nature Communications 13, 3412 (2022)]